



# CFSv2-based sub-seasonal precipitation and temperature forecast skill over the contiguous United States

Di Tian [1], Eric F. Wood [1], and Xing Yuan [2]

[1]Department of Civil and Environmental Engineering, Princeton University, Princeton, New Jersey 08544, USA
[2] RCE-TEA, Institute of Atmospheric Physics, Chinese Academy of Sciences, Beijing 100029, China

*Correspondence to*: Di Tian (dtian@princeton.edu)

**Abstract.** Forecasts from global seasonal climate forecast models can be potentially exploited for sub-seasonal forecasts of precipitation and 2-m temperature. The probabilistic sub-seasonal forecast skill of ten precipitation and temperature indices is investigated using the 28-years' hindcasts of the Climate Forecast System version 2 (CFSv2) over the contiguous United

States (CONUS). The forecast skill is highly dependent on the forecast indices, regions, seasons, leads, and methods. Indices characterizing mean precipitation and temperature as well as measuring frequency or duration of precipitation and temperature extremes for 7-, 14-, and 30-day forecasts were skillful depending on seasons, regions, and forecast leads. Forecasts for 7- and 14-day temperature indices showed skill even at weeks 3 and 4, and generally more skillful than precipitation indices. Overall, temperature indices showed higher skill than precipitation indices over the entire CONUS

region. While the forecast skill related to mean precipitation indices were low in summer over the CONUS, the number of rainy days, number of consecutive rainy days, and the number of consecutive dry days showed considerable high skill for the west coast region. The 30-day forecasts of precipitation and temperature indices calculated from the downscaled monthly CFSv2 forecasts are less skillful than those calculated from the daily CFSv2 forecasts, suggesting the potential usefulness of the CFSv2 daily forecasts for hydrological applications relative to the temporally disaggregated CFSv2 monthly forecasts.

While the presence of active Madden-Julian Oscillation (MJO) events improves CFSv2 weekly mean precipitation forecast skill over major areas of CONUS, MJO or El Niño Southern Oscillation did not have same strong effects on weekly mean temperature forecasts.

## 1. Introduction

Sub-seasonal (or intra-seasonal) timescale forecasts are typically between medium-range weather forecasts (1 or 2 weeks)
and seasonal climate predictions (1 to 12 months). The medium-range forecast is strongly influenced by atmospheric initial conditions (Vitart et al. 2008) while the seasonal climate forecast depend on slowly-evolving components of the earth system (e.g. sea surface temperature and soil moisture) (Troccoli 2010). However, since sub-seasonal timescale is usually beyond the memory of the atmospheric initial conditions (Vitart 2004) and too short to be strongly influenced by the variability of



the ocean, making skillful sub-seasonal forecasts is particularly difficult and thus far have received much less attention than the medium-range weather forecasts and seasonal climate forecasts.

Since many extreme events and management decisions fall into sub-seasonal timescales, accurate sub-seasonal forecast information will be central to the development of climate services and promise great socio-economic value (Vitart et al. 2012). For example, sub-seasonal forecast information can be used for reducing risks in management decision making and developing strategies for proactive natural disaster mitigation, which may be needed during drought and heat waves (Brunet et al. 2010; Vitart et al. 2012). Previous studies have evaluated the potential of sub-seasonal to seasonal forecasts for heat wave forecasting (e.g. Hudson et al. 2011a; White et al. 2014), hydrological forecasting (e.g. Orth and Seneviratne 2013; Yuan et al. 2014), water resources management (e.g. Sankarasubramanian et al. 2009), hydropower production management (e.g. Garcia-Morales and Dubus, 2007), and crop yield prediction (e.g. Hansen et al. 2006; Zinyengere et al. 2011). Due to the improvement of numerical models, prediction techniques, and computing resources, there is an increasing focus on sub-seasonal forecasts (e.g. Brunet et al. 2010; Hudson et al. 2011b; Hudson et al. 2013; Robertson et al. 2014; Toth et al. 2007; Vitart et al. 2008). Of many climatic variables, precipitation and 2-m temperature (hereafter temperature) are considered to be the most important factors to management decision making related to, for example: irrigation scheduling, urban water supply, cooling water related to thermal power generation, and hydropower operations. Many important sub-seasonal events including heat waves, cold waves, dry spells, and wet spells are directly derived from frequency, duration, and intensity of rainfall or hot (cold) temperatures. However, most of the studies for precipitation and temperature forecasts only focused on their mean or accumulated totals (e.g. Roundy et al. 2015). While several studies have been conducted to forecast the duration of high temperature days (i.e. heat waves) (e.g. Hudson et al. 2011a; Luo and Zhang 2012; White et al. 2014), there have been, thus far, no complete investigation of sub-seasonal forecasting capabilities for the other temperature and precipitation derivatives or indices that are directly associated with important events and decision making at sub-seasonal timescales.

Global climate models (GCMs) have become useful tools to make forecasts at multiple timescales. While the GCMs have demonstrated advanced configurations and realistic representations of the earth systems, the use of GCMs' predictions is still restricted by their coarse resolution and inherent systematic biases. To overcome these limitations, the GCMs' predictions at seasonal timescales are usually downscaled and bias corrected before being used in hydrological applications (e.g. Luo and Wood, 2008; Tian et al., 2014; Wood et al., 2002; Yuan et al., 2013). Despite the availability of daily or sub-daily forecast archives from the newly developed Climate Forecast System version 2 (CFSv2) by National Centers for Environmental Prediction (NCEP) (Saha et al. 2014) , temporal downscaling of the seasonal predictions is still routinely done from monthly to daily without using any of the daily forecast information (e.g. Yuan et al. 2013). At the sub-seasonal timescale, the usefulness of these daily or sub-daily precipitation or temperature forecasts compared to the monthly disaggregated forecasts has not been assessed. The CFSv2 has archived sub-daily hindcast datasets over almost 30 years, which provides an



opportunity to make such assessments. The CFSv2 has fully coupled atmospheric, oceanic, and land components of the earth systems and demonstrated the high performance for seasonal climate predictions when compared to other seasonal forecast models (Yuan et al. 2011). Since sub-seasonal precipitation or temperature forecasts are influenced jointly by the conditions of atmosphere, land, and ocean, the CFSv2 has great potential to make skillful precipitation or temperature forecasts at sub-seasonal timescales.

Besides GCMs, teleconnections between large-scale climate patterns and local weather events have also been used to develop sub-seasonal precipitation or temperature forecasts. Recent examples included sub-seasonal winter temperature forecasts in North America using Madden-Julian Oscillation (MJO) or El Niño Southern Oscillation (ENSO) conditions (Yao et al., 2011; Rodney et al., 2013; Johnson et al., 2013). In addition, Jones et al. (2011) found that the deterministic forecast skill of the CFSv1 for extreme precipitation in the contiguous United States (CONUS) during winter is higher when the MJO is active. With the updated version of CFS, the CFSv2 hindcasts allows re-examining this issue by assessing the influence of MJO or ENSO on the probabilistic temperature and precipitation forecast skill over the CONUS.

The aim of this study is to understand the capacities of using the CFSv2 for sub-seasonal precipitation and temperature predictions through a comprehensive evaluation of the precipitation and temperature hindcasts at sub-seasonal timescales. The assessment includes mean values of sub-seasonal predictions as well as related temperature and precipitation derivatives or indices at different forecast leads and scales. The downscaled CFSv2 monthly forecasts are compared with the native CFSv2 daily sub-seasonal forecasts. Furthermore, the influence of MJO or ENSO conditions on the CFSv2 probabilistic temperature and precipitation forecast skill is also assessed. This study will leverage sub-seasonal forecasting efforts and contribute to sectorial management decision making at sub-seasonal timescales.

## 2. Data and Methodology

The CFSv2 has the most updated data assimilation and forecast model components of the climate system and became operational at NCEP since March 2011 (Saha et al. 2014). The CFSv2 has archived three different types of hindcast (or reforecast): 6-hourly time series from 9-month runs, 45-day runs, and season runs. The 9-month runs are forecasts from initial conditions every 5 days apart (for all 6Z cycles on that day) for each calendar year over the 28-year period (1982-2009). The season runs are forecasts from every 0Z cycle over a 12-year period (1999-2010). The 45-day runs are forecasts from every 6Z, 12Z and 18Z cycle over the same 12-year period (1999-2010). The CFSv2 hindcast has a T126 spatial resolution (roughly 100 km) and includes several near surface variables at a 6-hourly temporal resolution. While the one season and 45-day reforecasts have initializations every single day and the relative new initial conditions can be incorporated in a large ensemble size for making a potentially skillful forecast, we chose to use the 9-month reforecast in that the 9-month reforecast covered much longer period than one season and 45-day reforecasts. With a longer period available, we would have larger sample size to ensure a more robust evaluation. In addition, the consideration of MJO and ENSO conditions





would result in even smaller sample sizes for evaluating the forecast skill conditioned on MJO and ENSO. Therefore, this study used the 9-month hindcast. The daily precipitation total was aggregated from the 6-hourly precipitation data; the daily mean temperature was obtained by averaging daily maximum and minimum temperature which were extracted from the 6-hourly maximum and minimum temperature. The ensemble members for each month were constructed in the same way as

CFSv2 producing monthly means hindcasts. For each year, the daily hindcast had 28 members in November and 24 members in other months with initial conditions of the 0000, 0600, 1200, and 1800 UTC cycles for every 5 days. For example, the 24 ensemble members for January are the four cycles for each of December 12[th], 17[th], 22[nd], and 27[th] and January 1[st] and 6[th]. The forecast validation dataset is from the North American Land Data Assimilation System version 2 (NLDAS-2; Xia et al., 2012). The forcing dataset of the NLDAS-2 merges a large observation-based and reanalysis data and is routinely used to

drive land surface models over the CONUS. It has $0.125^{\circ}$ (approximately 12 km) spatial resolution and hourly temporal resolution. The NLDAS-2 hourly precipitation (temperature) data were aggregated (averaged) into daily data.

Besides using CFSv2 daily hindcasts at its native spatial resolution (hereafter CFSv2 daily), the CFSv2 monthly hindcasts were also downscaled using the Bayesian merging (BM) method for hydrological applications (Luo et al., 2007). By comparing those two forecasts, it will help us understand the usefulness of the CFSv2 daily precipitation or temperature

forecasts for hydrological applications compared to the monthly disaggregated forecasts. The BM method both spatially and temporally downscaled the CFSv2 monthly hindcasts from its native spatial resolution into daily hindcasts at a $0.125^{\circ}$ spatial resolution for hydrological applications. The BM method updates an observational climatology based on the hindcasts using Bayesian theory and generate 20 daily ensemble members for each month using historical-analog criterion and random selection. For a more detailed description of the BM method, please see Luo et al. (2007) and Luo and Wood (2008).

Ensemble forecasts of precipitation and temperature indices at sub-seasonal timescale were calculated using daily forecasts from the raw CFSv2 and the BM downscaled CFSv2. For daily forecasts from the raw CFSv2, all precipitation and temperature indices were calculated at 7-, 14-, and 30-day forecast timescales in month 1. For daily forecasts from the BM downscaled CFSv2, the precipitation and temperature indices were only calculated at 30-day forecast timescales in month 1, since these forecasts were temporally disaggregated from monthly forecasts and it would be useful to look at the

performance of the raw CFSv2 daily forecast in comparison with the daily forecast disaggregated from the monthly forecast. Five precipitation indices were calculated, including mean precipitation (Pmean), mean precipitation over wet days (RainWet), number of rain days (RainDay), maximum rain day (wet spell) length (WetSpell), and maximum dry spell length (DrySpell) during the 30-, 14-, or 7-day periods in forecast month 1. Following Zhang (2011), a wet (dry) day is defined as days with precipitation above (below) 1 mm during the $n$-day period. The wet (dry) spell is defined as number of consecutive

wet (dry) days. Take the forecast for January 14-day WetSpell as an example, lead one (two) forecast is the number of consecutive rainy days from January 1 (15) to 14 (28). Similarly, five air temperature indices are calculated, including mean temperature (Tmean), number of high temperature days (HighDay), number of low temperature days (LowDay), maximum



number of consecutive high temperature days (CosHighD), maximum number of consecutive low temperature days (CosLowD) during the 30-, 14-, or 7-day periods in forecast month 1. As a way of defining heat (cold) wave (e.g. Spinoni et al. 2015), the threshold for high (low) temperature day is defined when the temperature is above (below) $90^{th}$ ($10^{th}$) percentile of climatological distribution of temperature during the *n*-day period for different months.

To validate the forecasts, the observed precipitation and temperature indices were also calculated using the NLDAS-2 daily precipitation and temperature data. The NLDAS-2 daily precipitation and temperature data were also upscaled using bin averaging in order to validate the raw CFSv2 forecasts. The percentiles of defining high (low) temperature were obtained separately from distributions of forecasts and observations. All ensemble forecasts including raw and BM downscaled CFSv2 forecasts were verified against the NLDAS-2. While the raw CFSv2 forecasts were evaluated at 30-day, 14-day, and

7-day time scales, the BM downscaled forecasts were only evaluated at 30-day time scale since they were disaggregated from monthly forecasts. Take raw CFSv2 forecasts for January as an example, there are 24 ensemble members for all 30-day, 14-day, and 7-day forecasts. The 24 member ensemble forecasts were considered as being initialized on the first day of the month regardless of which day the individual member of the forecasts was initialized. Those 24 member ensemble forecasts were verified for the common period of 1 Jan-30 Jan, 1 Jan-14 Jan, and 1 Jan-7 Jan, respectively. All ensemble

forecasts were converted into probabilistic forecasts in terciles. All observations were converted into dichotomous values of 1 or 0 in terciles. The terciles were defined separately based on the individual distributions of the observations and the forecasts (*x*), with *x*<1/3rd percentile for lower tercile, 1/3rd≤*x*≤2/3rd percentile for middle tercile, and *x*>2/3rd percentile for upper tercile.

  The probabilistic forecasts were evaluated using the Heidke Skill Score (HSS), a common performance metric used by the

Climate Prediction Center (CPC) (e.g. Johnson et al. 2013; Wilks 2011). The HSS assesses the proportion of correctly forecasted categories. The probabilistic forecast is assigned to three forecast categories (upper, middle, or lower tercile) based on the highest of the three forecast probabilities. The tercile category probabilities were obtained by counting the ensemble members in each of the three categories and then divided by the ensemble size. The HSS is expressed as:

$$HSS = \frac{(H - E)}{(T - E)} \times 100 \qquad (1)$$

The number of correctly forecasted categories is denoted as H. The random forecast, E, is the reference forecast, which is one-third of the total number of forecasts, T. The HSS ranges from-50 (no correct forecasts) to 100 (perfect forecasts), with 0 representing the same skill as randomly generated forecast, which in this case is the climatological forecast. The HSS above 0 indicates that the forecasts have skill. The HSS was assessed for each method (raw CFSv2 and BM), variable index, grid point, month, and forecast time. Since precipitation and temperature could be more predictable at larger scales (e.g. Luo and

Wood 2006; Roundy et al. 2015), it is worthwhile to also look at predictability of subseasonal forecasts averaged over a larger spatial domain. Therefore, each forecast was averaged over each of the nine National Climatic Data Center (NCDC)



climate regions as well as over the entire CONUS (Figure 1). The HSS of the average forecasts over each of those regions were evaluated subsequently.

[insert Figure 1 here]

The skill assessment of Pmean and Tmean was conducted not only for all forecasts but also for forecasts during active MJO, ENSO, or combination of the two.  MJO is the dominant mode of the sub-seasonal variability in the tropical atmosphere. The MJO index used in this study was from the Australian Bureau of Meteorology (http://cawcr.gov.au/staff/mwheeler/maproom/RMM/) for the period of 1982 to 2009. This index is defined by the two leading principal components (PCs) from an empirical orthogonal function analysis of the combined near-equatorially averaged 850-hPa zonal wind, 200-hPa zonal wind, and satellite-observed outgoing longwave radiation data (Wheeler and Hendon 2004). The pair of these two leading PC time series at a daily time step, called the Real-time Multivariate MJO series 1 (RMM1) and 2 (RMM2), define eight MJO phases and an MJO amplitude. There are a few different ways to define active MJO events. While the simplest criterion was defining MJO as RMM amplitude exceeding a certain threshold (e.g. Johnson et al. 2014), this criteria did not consider minimum duration and eastward propagation of MJO. Following L'Heureux and Higgins (2008), this study adopted a more rigorous definition of MJO: MJO days and events are identified using a pentad-averaged version of the Wheeler and Hendon RMM index subject to three major requirements: (i) the aptitude of the index must be greater than one for consecutive pentads; (ii) phases of the index must be in numerical order (i.e., phases 6, 7, 8, 1, 2); and (iii) events must last for more than 30 days (six consecutive pentads) and cannot be in a particular phase for more than 20 days (four pentads). When the amplitude of one pentad is slightly below one, they are still included as part of a larger MJO event. Similar definition was also widely adopted by other researchers such as Jones (2009) and Jones and Carvalho (2011). In this work, ENSO was defined using the same criteria as CPC (http://www.cpc.ncep.noaa.gov/products/analysis_monitoring/ensostuff/ensoyears.shtml). ENSO periods are based on a threshold of +/- 0.5 $^o$C for the Oceanic Niño Index (3 month running mean of SST anomalies in the Niño 3.4 region). ENSO periods of below and above normal SSTs are when the threshold is met for a minimum of 5 consecutive overlapping seasons.

## 3. Results

### 3.1 The 30-day forecast skill

Figures 2 show average HSS for 30-day forecasts of precipitation indices calculated from the CFSv2 daily at different locations over December-January-Februry (DJF) and June-July-August (JJA).

[insert Figure 2 here]

In DJF, the average skill of WetSpell over the CONUS is 34, which is much higher than the other indices; it showed high skill over major area of the CONUS including midwest and eastern US. The Pmean, RainDay, and DrySpell were skillful in





the southeast and the southwest but also revealed skill in other regions. RainWet showed minor skill over the entire region. The skill in JJA showed different spatial patterns with DJF. While the Pmean and RainWet showed modest forecast skill in JJA over the CONUS, the RainDay, WetSpell, and DrySpell all showed high skill in the west coast regions with the WetSpell showing some skill in the midwest and northeast. For the other seasons, on average, the forecast skill for

precipitation indices is between DJF and JJA for MAM but slightly lower than JJA for SON (not shown).

Spatial patterns in HSS are very different among the indices, particularly in July. We calculated the standard deviation (STD) for observed precipitation indices in July to further examine the interannual variability of those indices at each grid point over the space. To compare relative temporal variability in space, the STD was normalized spatially to a range of 0 to 1 using a feature scaling method:

$$STD^{'} = \frac{STD - \min(STD)}{\max(STD) - \min(STD)} \qquad (2)$$

Where STD is the standard deviation of time series for each grid point, min(STD) is the minimum STD over all grid points, max(STD) is the maximum STD over all grid points, and STD' is the normalized STD. Figure 3 shows normalized standard deviation of 30-day precipitation indices in July over 28-year period from 1982 to 2009 over the CONUS.

[insert Figure 3 here]

By comparing interannual variability (Figure 3) with the forecast skill over the space (Figure 2), we found that regions showing lower interannual variability usually have higher skill than the regions with higher interannual variability. For Pmean, the western CONUS showed relatively lower interannual variability and higher skill than the eastern CONUS; for RainDay, the western coastal areas showed much lower variability and higher skill than the other regions; for RainWet, all regions showed relatively equal variability and skills; for WetSpell, the southeastern CONUS showed higher interannual

variability and lower skill than other regions of the CONUS; for DrySpell, California and eastern CONUS showed relatively lower interannual variability and higher skill than the other areas.

Figures 4 show average HSS for 30-day forecasts of temperature indices calculated from the CFSv2 daily at different locations over DJF and JJA.

[insert Figure 4 here]

Overall, the temperature indices showed reasonably higher skill than the precipitation indices in both DJF and JJA. For DJF, Tmean showed moderate high skill in the Great Lakes area and eastern US; the HighDay, LowDay, CosHighD, and CosLowD were skillful over most areas of CONUS and the skill was particularly high for LowDay and CosLowD in the center or north of the midwest region. The forecast skill of temperature indices in DJF showed different spatial patterns with

JJA. Tmean and LowDay showed high skill over the west inland area. CosLowD is skillful over major area of the CONUS, particularly in the northeast. HighDay and CosHighD showed notable high skill around south of the central area. For the other seasons, on average, the forecast skill for temperature indices is between DJF and JJA for MAM but slightly lower than JJA for SON (not shown).

Figure 5 shows average HSS for 30-day forecasts of precipitation and temperature indices calculated from the CFSv2 daily or BM downscaled CFSv2 over 12 months for CONUS and its consistent NCDC climate regions.

[insert Figure 5 here]

The precipitation and temperature indices calculated from CFSv2 daily showed higher skill than BM for all regions. On average, the skill from the CFSv2 daily is approximately  20% higher than the skill from the BM, suggesting that the CFSv2

month-1 daily forecasts are potentially more useful than the temperally downscaled monthly forecasts for hydrological applications.

### 3.2 Fortnight and weekly forecast skill at different lead time

Figure 6 (Figure 7) shows average HSS of 14- and 7-day precipitation (temperature) indices forecasts from the raw CFSv2 over 12 months for CONUS and its consistent NCDC climate regions.

[insert Figure 6 here]
[insert Figure 7 here]

In general, the skill scores for precipitation indices  are reasonaly higher in the first two weeks than the second two weeks at both 14- and 7-day time scales, since first two weeks are within the range of weather forecast and are strongly influenced by the atmospheric initial conditions. While there are differences among regions, the skill scores for indices measuring

frequency or duration of precipitation (i.e. RainDay, WetSpell, and DrySpell) or temperature extremes (i.e. HighDay, LowDay, CosHighD, and CosLowD) were equally skillful as those measuring mean precipitation or temperature during the first two weeks. Temperature indices showed notably higher skill than any precipitation index, particularly in weeks 3 and 4.

### 3.3 Effects of MJO and ENSO

Figure 8 shows skill differences between Pmean or Tmean weeks 2-4 forecasts during active ENSO, MJO, or combined

active ENSO and MJO (MJO+ENSO) and the forecasts during all periods for CONUS and its consistent NCDC climate regions. The Pmean and Tmean forecasts are calculated from the CFSv2 daily. In general, weeks 3 and 4 Pmean forecasts perform better during active ENSO or MJO states, while Tmean forecasts do not perform better.



[insert Figure 8 here]

For precipitation, forecast skill is inconsistent for the active ENSO, MJO, or combined ENSO and MJO relative to all periods. There was a notable increase in skill when the forecasts were conditioned on active MJO for almost all regions, indicating the positive influence of MJO on the CFSv2 sub-seasonal precipitation forecasts. It is worthwhile to note that

forecasts conditioned on combined MJO and ENSO, and forecasts conditioned on MJO, showed similar level of positive skill with a few differences, which may due to the modulation effects of ENSO on MJO. For temperature, while the MJO, ENSO, or combined MJO and ENSO mostly showed positive effects on CFSv2 sub-seasonal temperature forecast skill for week 2 forecast, those influences became negative in most of the regions. We further examined differences between Pmean or Tmean average skill over weeks 2- 4 for forecasts during active MJO and for forecasts during all period at different

locations over the CONUS for DJF, MAM, JJA, and SON (Figures 9 and 10).

[insert Figure 9 here]
[insert Figure 10 here]

In general, MJO has strongly positive effects on CFSv2 subseasonal Pmean forecast skill over the CONUS; the effects on Tmean forecast skill is relatively weak and inconsistent among different regions. For precipitation, the influenced areas are

greater during DJF and MAM than during JJA and SON, with the NE and NW regions being consistantly influenced by MJO during four seasons. We further conducted statistical tests to compare whether precipitation forecast skills during active MJO, ENSO, or combined MJO and ENSO are greater than those during all period over the CONUS for DJF, MAM, JJA, and SON. We tested whether differences in mean HSS over the CONUS (averaged over 1024 grid points) are statistically significant at a 5% level. The student's t-test showed the forecast skills during active MJO or combined MJO and ENSO

were significantly greater than those during all period ($p<0.05$) over the CONUS for DJF, MAM, JJA, and SON; the forecast skills during active ENSO were significantly greater than those during all period over the CONUS for MAM.

## 4. Discussion

The CFSv2 subseasonal forecast skill is highly depending on forecasting indices, regions, seasons, leads, and methods. The sub-seasonal forecasts for indices characterizing mean precipitation and temperature as well as frequency or duration of

precipitation and temperature extremes showed skill in the first two weeks but no skill or modest skill for the second two weeks, since the first two weeks were within the range of medium-range weather forecasts. This finding is important since sub-seasonal forecasting information, particularly frequency or duration of precipitation and temperature extremes, is valuable to many decision makings and can thus be directly tailored to different application needs. For example, knowing RainDay, WetSpell, and DrySpell weeks in advance will help farmers make irrigation scheduling decisions, save water costs

and improve crop yields. Short-term planning of urban water supply could also benefit from this forecasting information since these precipitation and temperature indices are known to be directly related the urban water demand forecasting (e.g.





Donkor et al. 2012). As some temperature indices such as CosHighD and CosLowD were used to characterize hot/cold waves, forecasting information of these indices would also be useful for developing strategies for proactive disaster mitigation (e.g. frost damage to crops).

The spatial and temporal downscaled CFSv2 monthly data using BM method was compared with the CFSv2 daily data for sub-seasonal forecasts at native resolutions. It is worth to note that the skill scores only reflect the performance of the forecast anomalies since the terciles were defined individually for the forecasts and observations. The BM method, while also serving as a bias correction method in previous hydrological forecasting studies (e.g. Yuan et al. 2013), mainly played the role of spatial and temporal downscaling in this study. For forecasting 30-day precipitation and temperature indices, since the precipitation and temperature indices calculated from CFSv2 daily have slightly higher skill than the BM in most cases, the comparison of these two methods implies that daily forecasts from the CFSv2 are potentially more useful than those disaggregated from the monthly forecasts. Thus, the CFSv2 daily forecast information should be used in application studies of sub-seasonal hydrological forecasts in contrast to temporal disaggregation of the monthly forecast.

This study demonstrated the CFSv2 sub-seasonal forecast skill varies with space and time. These results identify seasons and regions where there is the potential for skillful sub-seasonal predictions for certain precipitation and temperature indices. For example, water managers in California trying to predict WetSpell and DrySpell have confidence to use the forecasts from CFSv2 during summer seasons, while a decision maker in the southeast may benefit little by using such information.

Sub-seasonal forecast skill can be further improved by understanding the attribution of the skill. This study took a first look at the effects of MJO and ENSO on the CFSv2 sub-seasonal forecast skill. It was found that the presence of an active MJO improves weeks-2 to -4 probabilistic CFSv2-based forecast performance of precipitation over major areas of CONUS. This finding corresponds to the study of Jones et al. (2011), who found improved deterministic CFSv1 forecast skill of extreme precipitation during active MJO. We also compared the regions of improved skill associated with the MJO in this study (Figure 13) with the regions shown in Figure 5 in Jones et al. (2011). While there were spatial differences, the regions of improved skill associated with the MJO commonly occurred for the western coast of the United States (US). This result is consistent with current knowledge of the observed influence of the MJO on precipitation events along the US west coast, which can be viewed at the NOAA CPC website (http://www.cpc.ncep.noaa.gov), under the MJO section. Forecast skill of precipitation and temperature are inherently associated with the capacity of CFSv2 in forecasting MJO. The CFSv2 has shown useful MJO prediction skill out to 3 weeks (Wang et al. 2014). Improvements of the representation of the MJO in CFSv2 will likely further extend the forecast skill of precipitation and temperature. Furthermore, related studies have developed statistical forecasting models at sub-seasonal timescale using teleconnections of MJO and ENSO phases and local weather (e.g. Johnson et al. 2013). These statistical models could be potentially combined with CFSv2 forecasts to further improve the sub-seasonal forecast skill.





It is opportune to note some future directions of this work. Forecast skill could be further improved by having a larger ensemble size. For future work, when one season or 45-day CFSv2 reforecasts are available over a longer period, we would choose to use those datasets instead of 9-month reforecasts in order to incorporate a large ensemble size for making a potentially more skillful forecast. Another approach to further improve the sub-seasonal forecast skill is through multi-model

ensembles. The multi-model ensemble forecasts combine multiple seasonal forecast models and often have higher skill than any individual model, since it has an increased ensemble size and a wider spectrum of possible forecasts that takes into account model uncertainty due to differences in model configuration and physics (e.g. Hagedorn et al. 2005). Here we highlighted two important endeavors: the North American Multi-Model Ensemble (NMME-2) system (Kirtman et al. 2013) is exploring sub-seasonal forecast in their next phase; the World Meteorological Organization (WMO) subeasonal to

seasonal (S2S) prediction project (http://www.s2sprediction.net/) is archiving hindcast and real-time forecasts from a range of model systems. All of those efforts can facilitate subseasonal multi-model ensemble prediction and model inter-comparison studies. Furthermore, this study focused on evaluation of the capacities of CFSv2 sub-seasonal precipitation and temperature forecasts. The CFSv2 sub-seasonal precipitation and temperature forecasts can be used for subsequent application studies related to areas such as hydrology and agriculture. For example, flash drought refers to a sudden onset of

high temperatures and decreases of soil moisture and is a disastrous event at sub-seasonal timescale (e.g. Mo and Lettenmaier 2015). Sub-seasonal forecasting of flash drought will help decision makers develop mitigation strategies. CFSv2 sub-seasonal precipitation and temperature forecasts can be used to drive land surface hydrological models to forecast soil moisture and evapotranspiration and consequently improve flash drought forecasts.

## 5. Conclusion

In this study, we have assessed CFSv2 probabilistic sub-seasonal forecasts of precipitation and temperature indices over the CONUS. The probabilistic sub-seasonal forecast skill is highly dependent on forecasting indices, regions, seasons, and methods. Indices characterizing mean precipitation and temperature as well as measuring frequency or duration of precipitation and temperature extremes for 7-, 14-, and 30-day forecasts were skillful depending on seasons and regions. Forecasts for 7- and 14-day temperature indices even showed skill at weeks 3 and 4, and generally more skillful than

precipitation indices. Forecasts of 30-day temperature and precipitation indices calculated from the daily forecasts BM downscaled from the monthly forecasts mostly showed lower skill compared to those calculated from the CFSv2 daily forecasts, indicating the potential usefulness of the CFSv2 daily forecasts for hydrological applications relative to the temporally disaggregated CFSv2 monthly forecasts. The presence of an active MJO improves weeks 2 to 4 probabilistic forecast performance of precipitation over major areas of CONUS in the CFSv2 system. The sub-seasonal forecast skill of

precipitation and temperature could be further improved through combining with teleconnection-based statistical sub-seasonal forecasting models or multi-model ensemble.

## Acknowledgments



This research was supported by the NOAA Climate Program through the grant NA12OAR4310090 entitled "A US National Multi-Model Ensemble ISI Prediction System", which is gratefully acknowledged. The authors would like to thank Dr. Michelle L'Heureux of the NOAA Climate Prediction Center for sharing daily MJO events data. The authors acklowledge PICSciE/OIT at Princeton University for the supercomputing support.

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





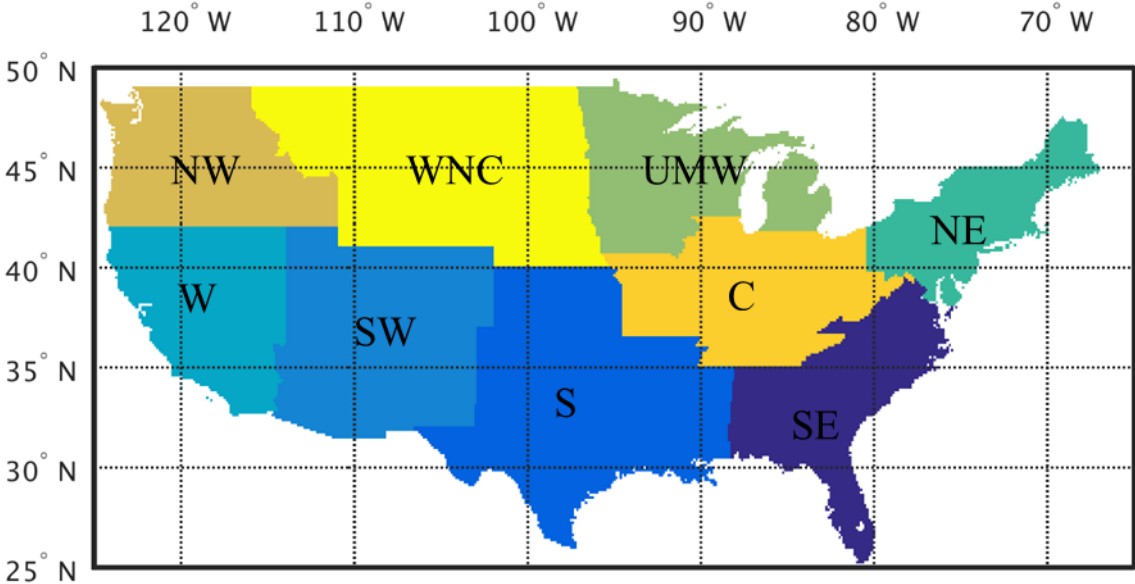

**Figure 1. NCDC climate regions (described in Section 2) used as area averaging domains for raw and BM downscaled CFSv2 forecasts. Regions are named as follows: Northwest (NW), West (W), Southwest (SW), West North Central (WNC), South (S), Upper Midwest (UMW), Central (C), Southeast (SE), and Northeast (NE).**





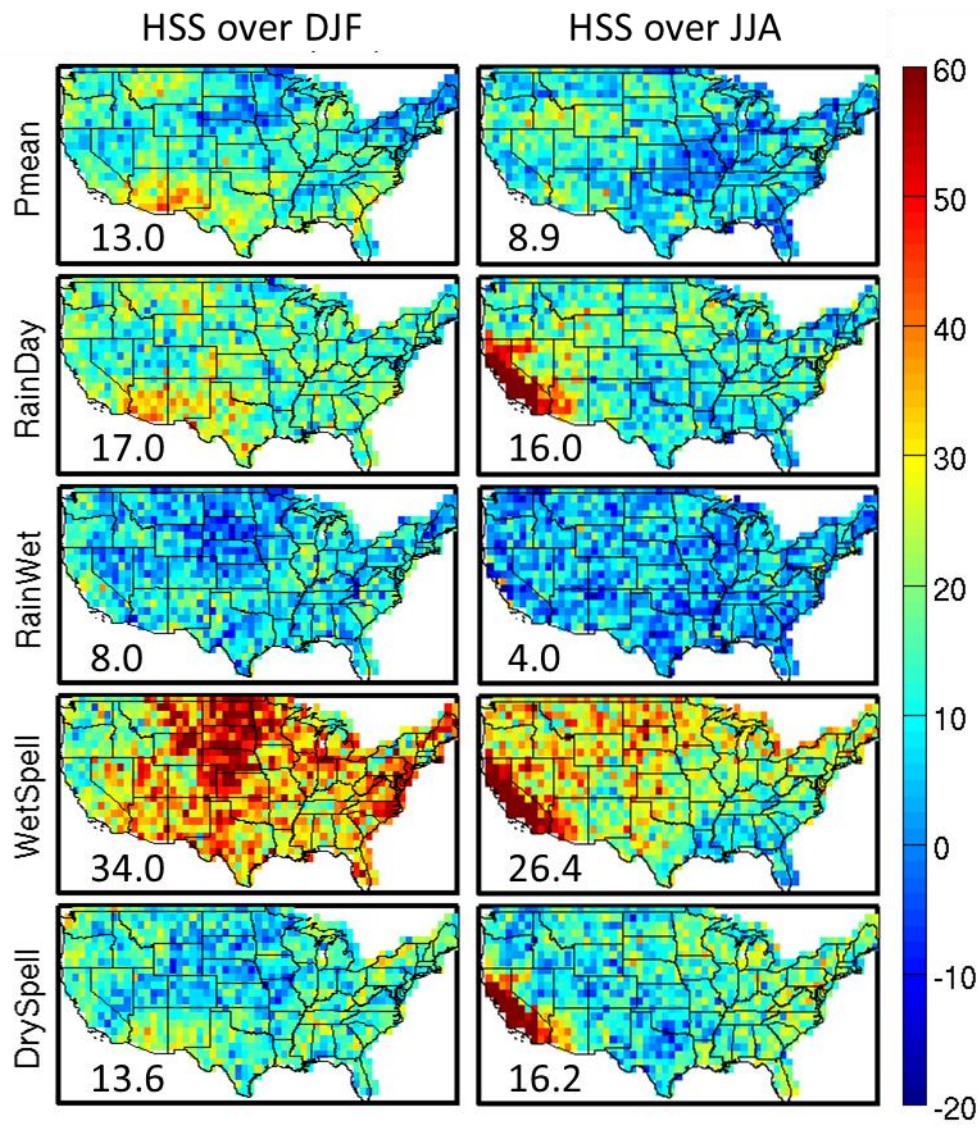

**Figure 2.** HSS of 30-day (from top to bottom columns) Pmean, WetRain, RainDay, WetSpell, and DrySpell from (from left to right rows) the Raw and BM over DJF (left) and JJA (right). The number in the bottom left is the overall average.




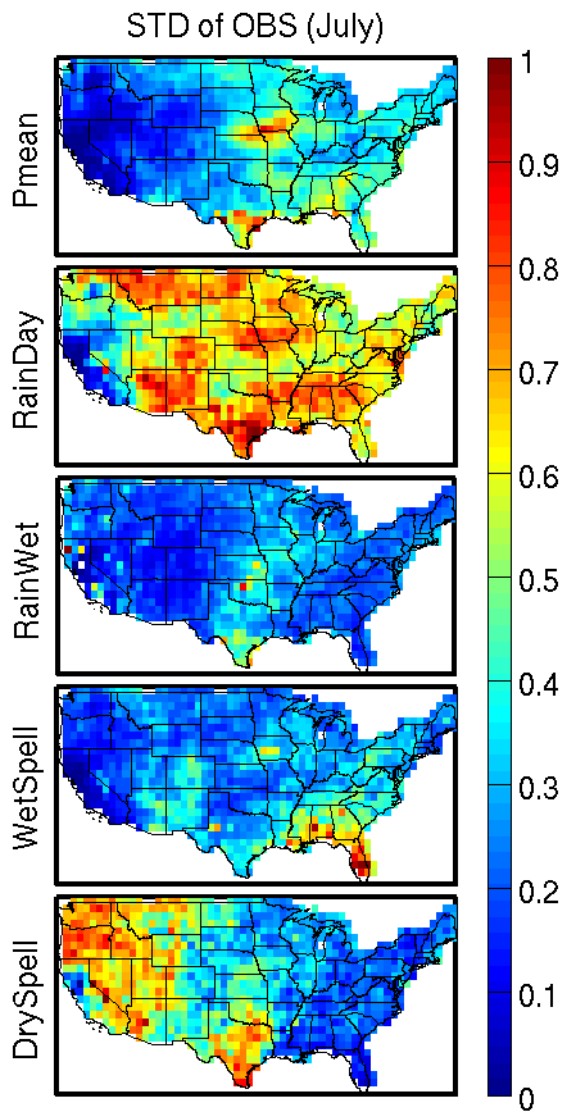

**Figure 3. Spatially normalized standard deviation of observed 30-day precipitation indices in July over 28-year period from 1982 to 2009**





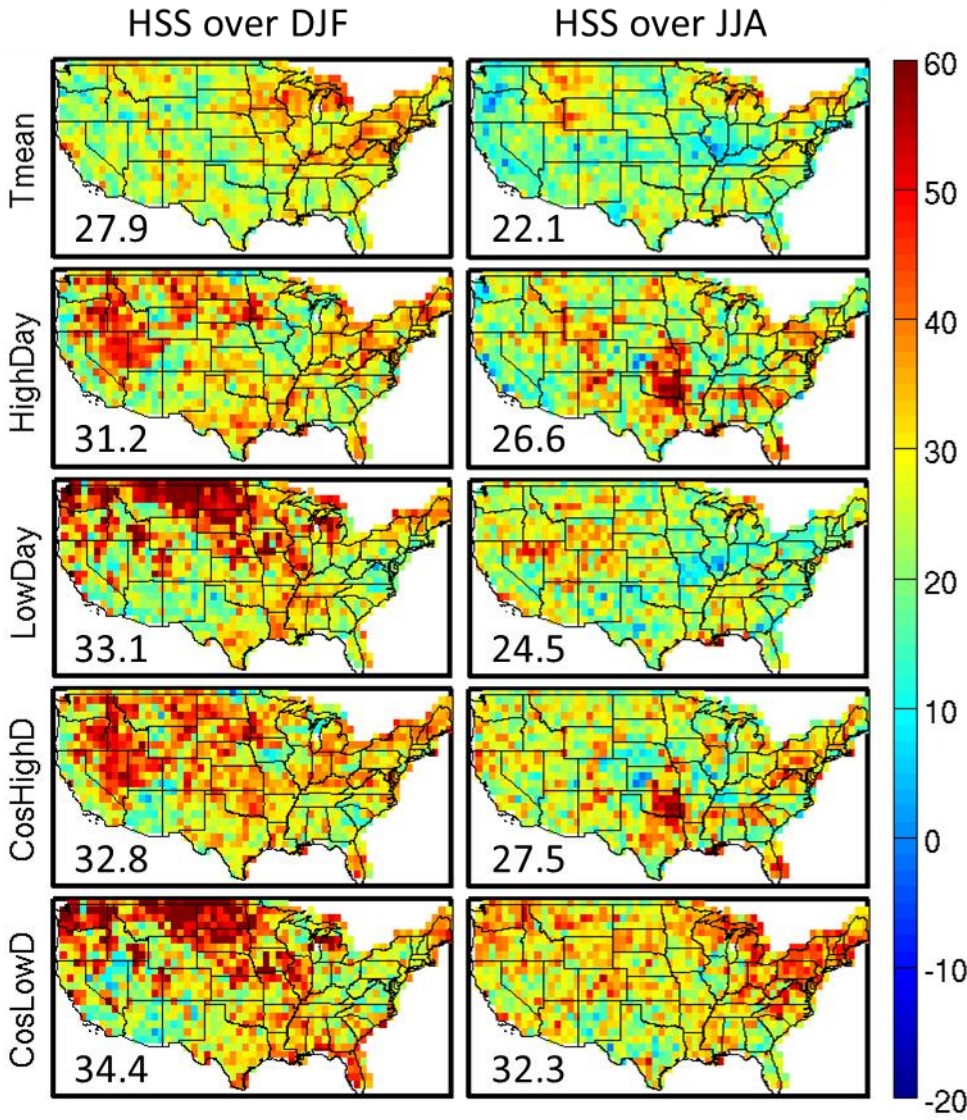

**Figure 4. HSS of 30-day (from top to bottom columns) Tmean, HighDay, LowDay, CosHighD, and CosLowD from (from left to right rows) the Raw and BM over DJF (left) and JJA (right). The number in the bottom left is the overall average.**





**Figure 5. HSS of 30-day precipitation and temperature indices calculated from the CFSv2 daily and BM for CONUS and its consistent NCDC climate regions. The red line is the overall average.**





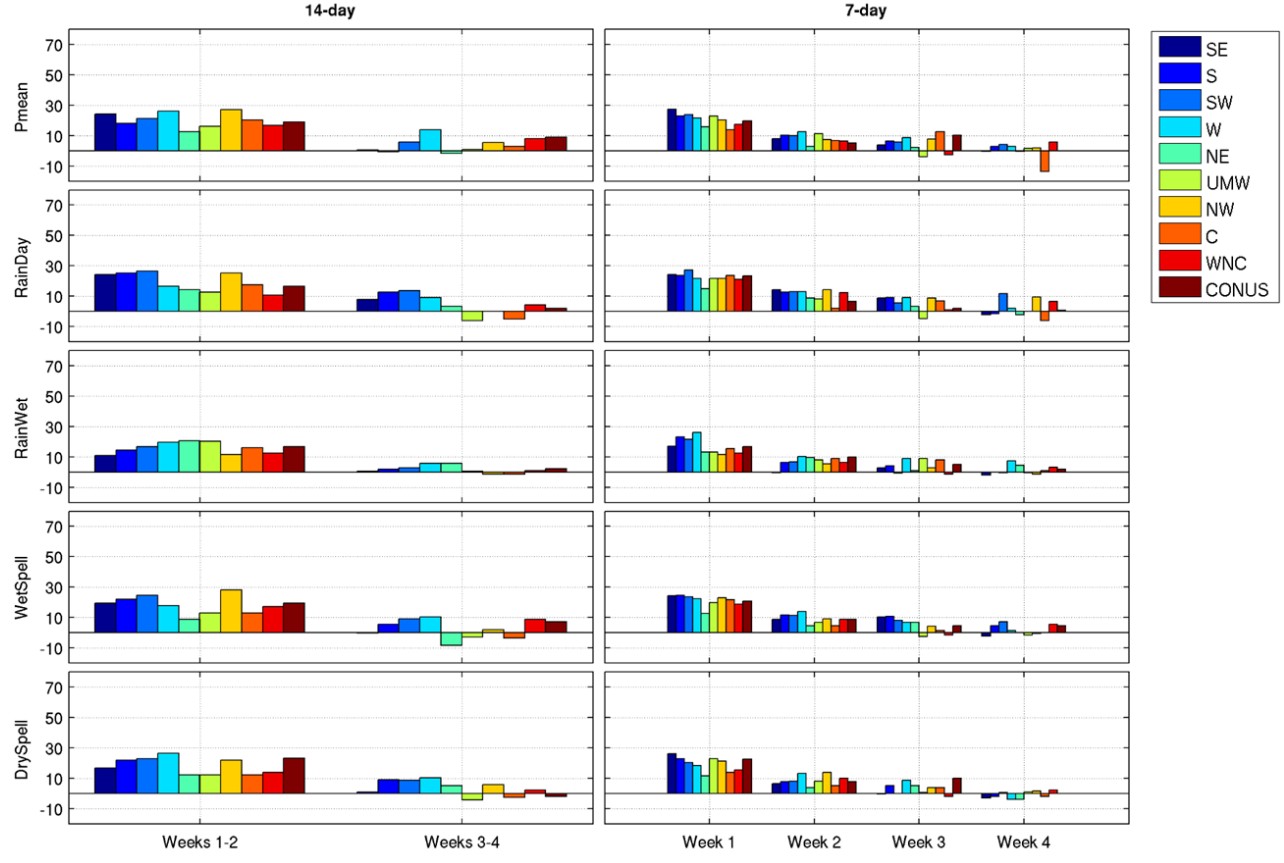

**Figure 6. Overall mean HSS of 14- and 7-day (from top to bottom rows) Pmean, WetRain, RainDay, WetSpell, and DrySpell from the Raw for CONUS and its consistent NCDC climate regions**





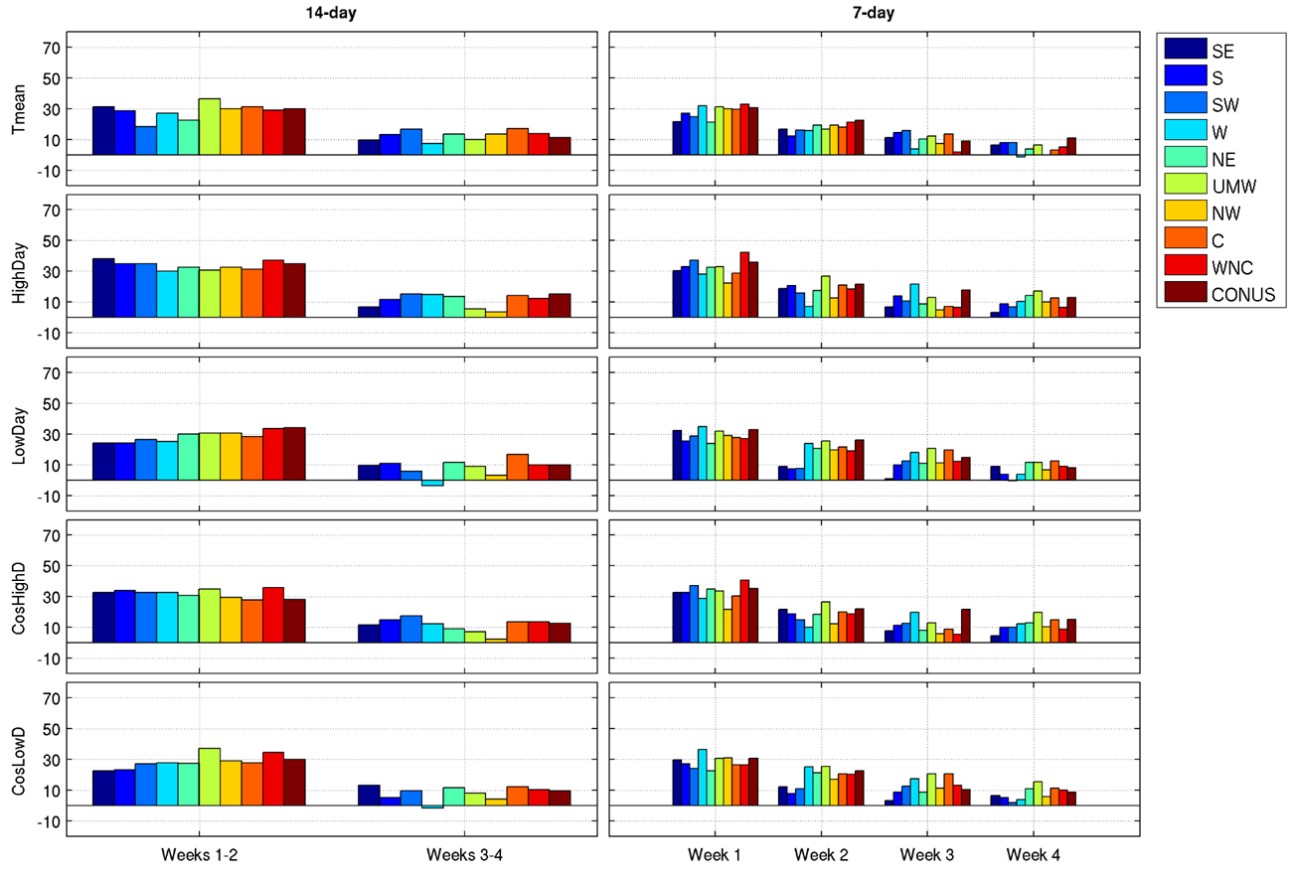

**Figure 7. Overall mean of HSS of 14- and 7-day (from top to bottom rows) Tmean, HighDay, LowDay, CosHighD, CosLowD from the Raw for CONUS and its consistent NCDC climate regions**





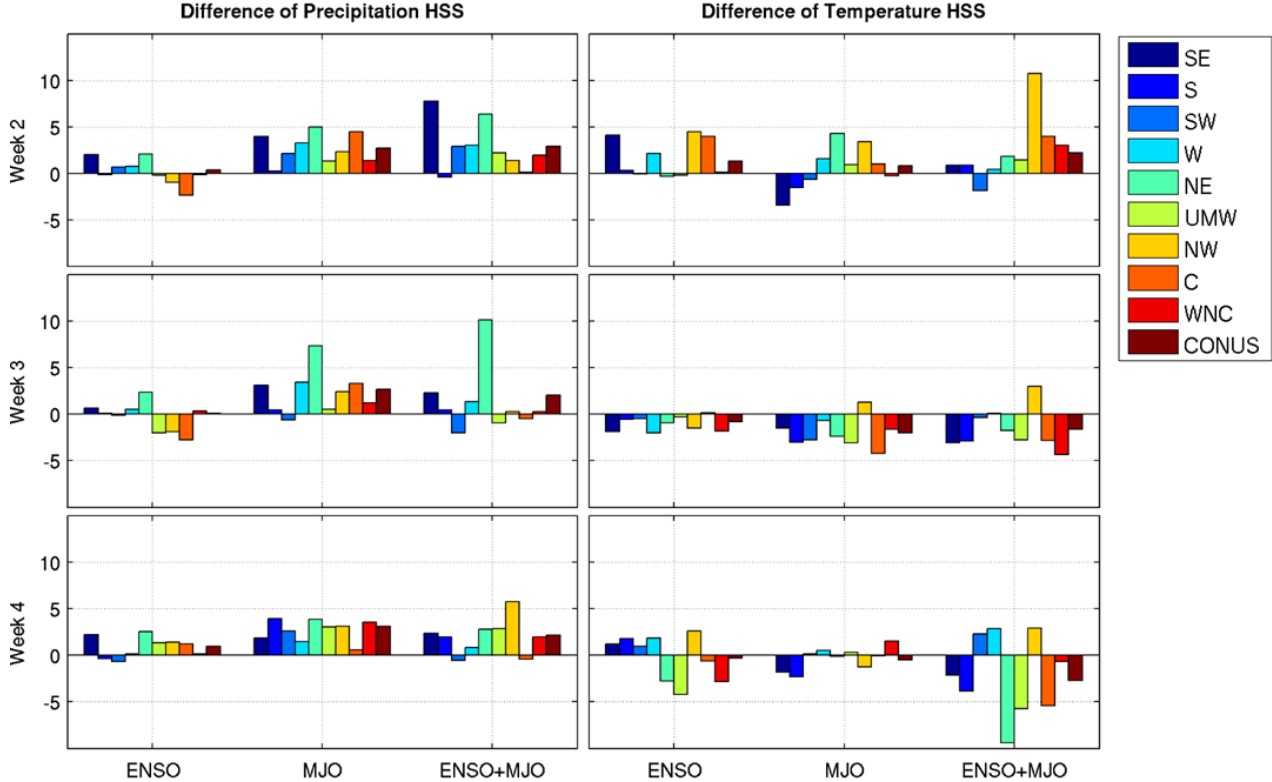

**Figure 8. HSS differences between Pmean (left column) or Tmean (right column) weeks 2-4 forecasts during active ENSO, MJO, or combined active ENSO and MJO (MJO+ENSO) and the forecasts during all periods for CONUS and its consistent NCDC climate regions. Positive values indicate more skillful forecasts for the active MJO, ENSO, or ENSO+MJO.**





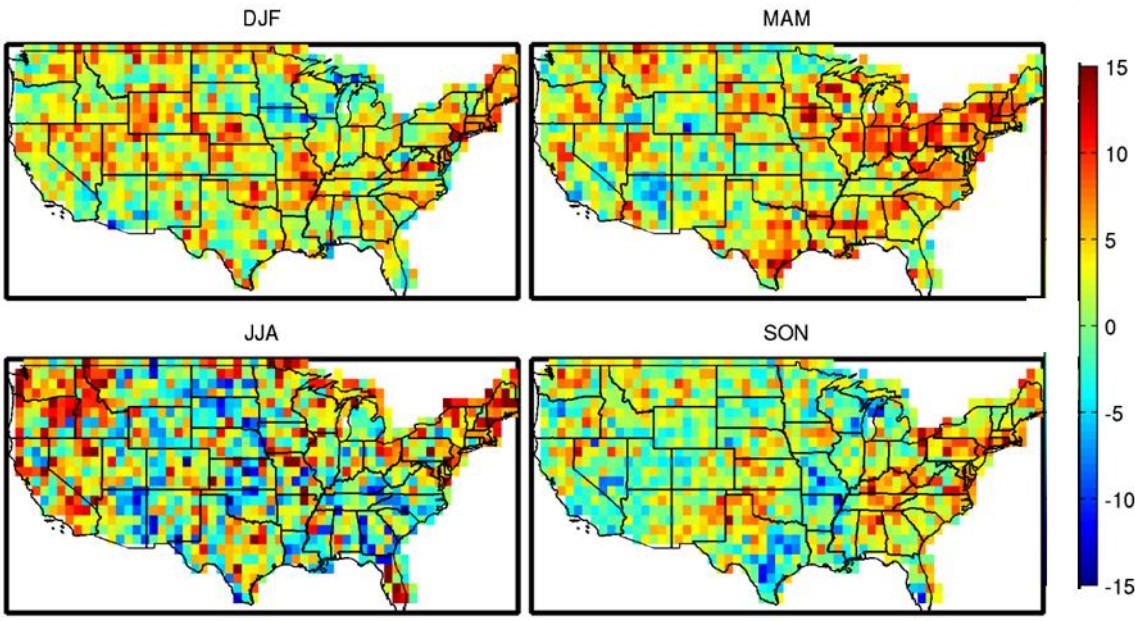

**Figure 9. Differences between Pmean average HSS over weeks 2-4 for forecasts during active MJO and for forecasts during all period at different locations over the CONUS for DJF, MAM, JJA, and SON.**





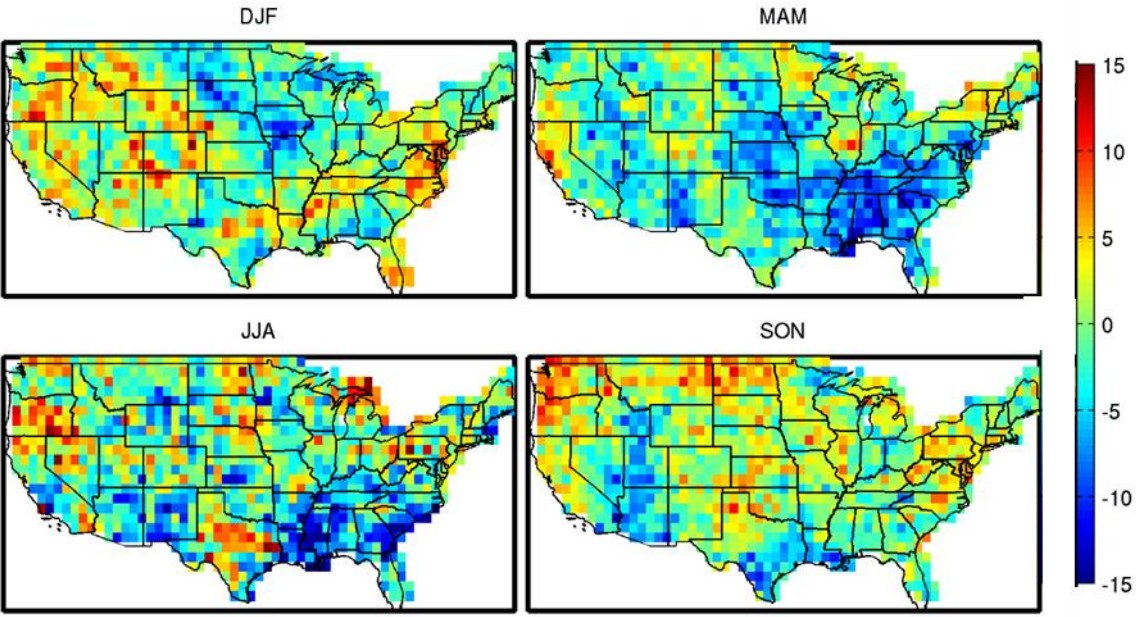

**Figure 10. Same as in Figure 9, but for Tmean.**