# Peer review of "CFSv2-based sub-seasonal precipitation and temperature forecast skill over the contiguous United States"

_Hydrology and Earth System Sciences, 2016_

## Referee Comment (RC1) · Anonymous Referee #1 · 17 Aug 2016

General comment: I think this paper has merit for publication but needs a little more work before acceptance. Most notable is the grammar and syntax which makes the paper difficult to read. Moreover, there are lots of different combinations of forecast times and forecast skill evaluations for different models. I found that following the different periods was confusing. Perhaps add a table with the appropriate information or structure the workflow differently. I think that the results and discussion are in line with what the paper aims to show. I did not find any methodological fault, although I must admit that the subject matter is not my main expertise.

Other comments and suggestions:

Abstract: I think the first sentence is either too vague or too direct. Perhaps start with

something like: "This paper explores the possibility of exploiting forecasts from global seasonal climate forecast models for sub-seasonal forecasts of precipitation and 2-m temperature". The current wording seems like a statement: "... forecast models can be...", but it is a vague statement because of the word "potentially".

Page2, line 12: References are not ordered properly; please revise (all text).

Page 2, line 12-15: This sentence is not clear, please revise.

Page 2, line 20: "... there have has been"

Page 2, line 23-25: Please explain how GCM outputs can be used for daily or short-term forecasts seeing as they are uncorrelated to current meteorological conditions.

Page 2, line28: The link between GCMs and the CFSv2 is not clear. Is the CFSv2 a GCM? Please indicate that it is a reforecast product based on reanalysis (If i understood correctly).

Page 3, line 2: "... demonstrated the high performance...": delete "the".

Page 3 lines 18-19: This sentence is not clear and does not add much to the paper. I suggest modifying it by giving it more substance. "Leverage forecasting efforts" and "contribute to sectorial management decision making" are both very vague objectives.

Page 3, line 22: "... forecast model components of the climate system... ".not clear what this means. Of all existing models?

Page 3, lines 29-33 : Check grammar here (everywhere, but particularly here). It is difficult to read.

Page 4, line 14: Replace the sentence with something like: "Comparing those two forecasts will help understand..."

Page 4, line 30-31: This should refer to a figure or a table somehow. We cannot follow the given example because of the lack of a reference.

[Figure]

Page 5, line 15: Merge to make a more fluid sentence? (e.g. All forecasts... and all observations...)

Page 6, lines 14-19: This section is suspiciously similar to the text in L'heureux and Higgins. Please reword or cite directly.

Figure 6-7: It is not clear to me why the score is higher for the 14-day (week 1-2) than for weeks 1 and 2 taken individually.

Page 10, line 25: Reference to nonexistent figure 13.

───────────────

---

## Referee Comment (RC2) · Anonymous Referee #2 · 26 Aug 2016

Summary & Referee Comment:

General Comment: This paper is interesting and provides information about the CFSv2 model that will be useful to the scientific community. The paper presents intriguing findings, but some are hard to decipher due to grammatical and syntax errors. I found the paper to be hard to follow in places due to the unclear description of different models and time steps. The "Data and Methodology" section which describes the models/time steps analyzed was particularly difficult to follow; some of the paragraphs could be split and the writing could be more concise. Overall, the scientific approach and methods seem adequate, though I am new to this subject. The paper also presents clear conclusions that are a substantial contribution to scientific progress.

Below are other specific comments and suggestions:

Pg 1, line 13: Missing a word - "and generally are more skillful"

Pg 1, line 16: Remove "the" before "number of consecutive dry days..."

Pg 1, line 26: Add a comma after the reference.

Pg 2, lines 13-19: To make the aims of this study more clear, it would be good to list or number them. There are 3 in the results section. It would be nice to see the same differentiation here.

Pg 3, lines 25-27: You refer to 0Z cycle (and different numbers of cycles). I am not familiar with this terminology, and am therefore confused when reading these sentences.

Pg 3-4, lines 22-33, 1-11: This paragraph is confusing due to the large amount of information presented. The paragraph could be split into multiple paragraphs and reworded to clarify which time steps and models are being referred to.

Pg 4, line 26: This could be the start of a new paragraph to separate the two different ideas.

Pg 5, line11: Replace comma with a semicolon or separate into two different sentences.

Pg 8, line 23: Bold the title.

Pg 9, lines 26-28: This sentence is confusing and could be reworded to improve clarity.

Pg 9, line 31: It is unclear which precipitation and temperature indices are being referred to with the word "these."

Pg 10, lines 6-12: These sentences could be rewritten to be more concise.

Pg 10, line 22: Figure 13 isn't used in the paper.

Pg 11, line 9: "Subseasonal" is spelled wrong and is also written differently than the

other instances where the spelling was "sub-seasonal." This may be an issue throughout the paper and should be checked.

---

## Referee Comment (RC3) · Anonymous Referee #3 · 10 Sep 2016

General Comments and Suggestions:

The work presented here is interesting, and presents an opportunity for researchers to utilize sub-seasonal forecasts from the CFSv2 model. However, this manuscript would benefit from a thorough proofreading, as the authors' writing is difficult to follow throughout much of the paper. Overall, I believe the manuscript can be made more concise and direct, which I believe will improve the readability of the paper. I have noted some specific instances below, but I am sure it is not a comprehensive list of all the improvements that could be made to the manuscript.

On page 2, the authors state that, "...many extreme events and management decisions fall into sub-seasonal timescales..." I suppose I could use some examples. As the

paper is currently, "sub-seasonal timescales" are on the order of 3-4 weeks. When I think of extreme events, I think of shorter term events such as flash flooding, tornadoes, extreme hail, wind, etc. . ., or I think of more persistent events such as drought or persistent floods. Many of the examples the authors provide in the second paragraph tend to be on the order of the seasonal timescale as defined by the authors on page 1. It would be beneficial if the authors could state what extreme events they specifically have in mind to be addressed by this research. Some of this is touched on near the end of the Discussion section by the authors, but could be more succinctly stated earlier.

At the end of the second paragraph on page 2, the authors state that the, ". . .derivitives or indices are directly associated with important events and decision making. . .". Similar to the previous comment, I do not believe the manuscript currently addresses this point.

Beginning on page 3 and continuing to page 4, the first paragraph of the Data and Methodology section is very confusing. A table comparing the differences between the 9-month, 45-day, and season runs would be beneficial. As it stands now, I believe this section assumes too much of the reader to interpret how NCEP runs the CFSv2 model.

On page 4, the authors discuss an example and reference a "lead one" forecast. It appears they are referencing the 14-day forecast, but it's not clear.

Equation 1 is not clear. The variable E is described as 1/3 the total number of forecasts, T. Why wouldn't the denominator simply be $(T - T/3)$? It sounds as if there is more to the variable E than is described.

With regards to the discussion of Figure 2 that starts near the end of page 6 and continues on to page 7, I am not sure why the authors do not include similar figures for MAM or SON, but do discuss the results. Wouldn't it be beneficial to include those figures? This comment applies to Figure 4 as well.

Similarly, I'm not sure why the authors stress emphasizing the month of July in Figure

3. The authors state there is some difference in monthly spatial patterns, but without the other months, I do not have the proper context for the figure.

With regards to section 3.3, it would be useful to have some sort of table or reference to see what periods of MJO and/or ENSO activity are being analyzed. The number of events considered could be limited enough that the HSS could be somewhat skewed. It would also be helpful if the authors discussed more clearly the impacts of active MJO and ENSO compared to the combined impacts of MJO and ENSO events. I think the authors begin to discuss this in the second paragraph on page 9, but do not offer enough insight on the particular point.

The first full paragraph on page 10, describing the role of the BM method is not clear. I would recommend the authors explain this conclusion more clearly, or simply remove the paragraph.

On page 11, the authors state that forecast skill could be improved by simply having a larger ensemble. I am not convinced of that; a larger ensemble may not necessarily add useful information. It may be more appropriate to state that a sensitivity study on ensemble size could be performed to see if a larger ensemble does improve forecast skill.

Page 1, lines 27-28: I'm not sure what is meant by, "...sub-seasonal timescale is beyond the memory of the atmospheric initial conditions..."

Page 5, the NCDC has since been renamed the National Centers for Environmental Information (NCEI)

Page 6, line 26 should read "Figure 2 shows"

There is no legend for Figure 5, so I am unsure which regions match to each particular color.

Page 8, line 10: "temperally" should be "temporally"

Page 8, line 17: "reasonaly" should be "reasonably"

Page 9, line 23: "depending" should be "dependent"

Page 10, line 4, "spatial" and "temporal" should be "spatially" and "temporally"

Page 10, line 5: "to note" should be "noting"

Page 10, line 23: There is no Figure 13 included in the manuscript. To this point, referencing a figure in another article (Jones et al. 2011) is a bit confusing. I think I would just note how the results of this study compare to Jones et al., rather than to a specific figure.

Page 11, line 8, "highlighted" should be "highlight"

Page 11, line 9, "subeasonal" should be "subseasonal"

---

## Author Comment (AC1) · 16 Oct 2016

Response to Reviewer #1:

General comment: I think this paper has merit for publication but needs a little more work before acceptance. Most notable is the grammar and syntax which makes the paper difficult to read. Moreover, there are lots of different combinations of forecast times and forecast skill evaluations for different models. I found that following the different periods was confusing. Perhaps add a table with the appropriate information or structure the workflow differently. I think that the results and discussion are in line with what the paper aims to show. I did not find any methodological fault, although I must admit that the subject matter is not my main expertise.

*RESPONSE: Thank you very much for your time and your insightful review. We have improved the grammar and syntax throughout the manuscript. We added tables with appropriate information to clarify the description of data and method. We have carefully revised the manuscript in order to include your comments. We believe that this manuscript is substantially improved as a result of the revision.*

Other comments and suggestions:

Abstract: I think the first sentence is either too vague or too direct. Perhaps start with something like: "This paper explores the possibility of exploiting forecasts from global seasonal climate forecast models for sub-seasonal forecasts of precipitation and 2-m temperature". The current wording seems like a statement: "... forecast models can be...", but it is a vague statement because of the word "potentially".

*RESPONSE: This sentence has been revised as suggested.*

Page2, line 12: References are not ordered properly; please revise (all text).

*RESPONSE: The orders of all the references have been adjusted throughout the text.*

Page 2, line 12-15: This sentence is not clear, please revise.

*RESPONSE: This sentence has been clarified as requested:*

*"Precipitation and 2-m temperature (hereafter temperature) are considered to be two of the most important climate variables that significantly influence irrigation scheduling, urban water supply, cooling water related to thermal power generation, and hydropower operations, etc."*

Page 2, line 20: "... there have has been"

*RESPONSE: It has been revised as requested.*

Page 2, line 23-25: Please explain how GCM outputs can be used for daily or short-term forecasts seeing as they are uncorrelated to current meteorological conditions.

*RESPONSE: Thanks for pointing this out. This is an overstatement. We have changed the sentence to "Coupled Atmosphere-Ocean General Circulation Models (GCMs) are used to make forecasts at multiple timescales."*

Page 2, line28: The link between GCMs and the CFSv2 is not clear. Is the CFSv2 a GCM? Please indicate that it is a reforecast product based on reanalysis (If i understood correctly).

*RESPONSE: The CFSv2 is a GCM in that it is a fully coupled ocean-land-atmospheric model, developed by NOAA for dynamical seasonal forecasting, and has archived a retrospective forecast product. It has been clarified in the revised manuscript.*

Page 3, line 2: "... demonstrated the high performance...": delete "the".

*RESPONSE: It has been deleted as suggested.*

Page 3 lines 18-19: This sentence is not clear and does not add much to the paper. I suggest modifying it by giving it more substance. "Leverage forecasting efforts" and "contribute to sectorial management decision making" are both very vague objectives.

*RESPONSE: We agree that this sentence is redundant and does not add much to the paper. We have deleted it in the revised manuscript.*

Page 3, line 22: "... forecast model components of the climate system... ".not clear what this means. Of all existing models?

*RESPONSE: This sentence has been changed to: "The CFSv2 has the most updated data assimilation and modeling system. It became operational at NCEP since March 2011."*

Page 3, lines 29-33: Check grammar here (everywhere, but particularly here). It is difficult to read.

*RESPONSE: These sentences have been changed to: "The one season and 45-day reforecasts are initialized each day so that relatively new initial conditions can be incorporated into a large ensemble for making a potentially skillful forecast over this shorter forecast period. Nevertheless, we chose to use the 9-month reforecast. This is because the 9-month reforecast covered much longer period (1982-2009) than the one season and 45-day reforecasts (1999-2010), which ensures a larger sample size for a more robust evaluation."*

*We have also proofed all the grammars throughout the text.*

Page 4, line 14: Replace the sentence with something like: "Comparing those two forecasts will help understand..."

*RESPONSE: We have changed the sentence to: "Comparing those two forecasts will help understand the usefulness of the CFSv2 daily precipitation or temperature forecasts for hydrological applications compared to the monthly disaggregated forecasts."*

Page 4, line 30-31: This should refer to a figure or a table somehow. We cannot follow the given example because of the lack of a reference.

*RESPONSE: We have added a table to explain this example.*

Page 5, line 15: Merge to make a more fluid sentence? (e.g. All forecasts... and all observations...)

*RESPONSE: This sentence has been changed to "All ensemble forecasts were converted into probabilistic forecasts in terciles with all observations converted into dichotomous values of 1 or 0."*

Page 6, lines 14-19: This section is suspiciously similar to the text in L'heureux and Higgins. Please reword or cite directly.

*RESPONSE: We reduced this section and made a more direct reference to L'Heureux and Higgins (2008).*

Figure 6-7: It is not clear to me why the score is higher for the 14-day (week 1-2) than for weeks 1 and 2 taken individually.

*RESPONSE: We have clarified this in the revised manuscript and added these sentences: "It is worth noting that the skill is higher for the 14-day forecast at the first lead than for 7-day forecast in weeks 1 and 2 taken individually. The improved forecast skill indicates that the temporal noise in predictions can be reduced through averaging, as noted by Roundy et al. (2015)."*

Page 10, line 25: Reference to nonexistent figure 13.

*RESPONSE: It should be Figure 9 (Figure 12 in the revised manuscript) instead of 13. We have revised that in the manuscript.*

---

## Author Comment (AC2) · 16 Oct 2016

Response to Reviewer #2:

General Comment: This paper is interesting and provides information about the CFSv2 model that will be useful to the scientific community. The paper presents intriguing findings, but some are hard to decipher due to grammatical and syntax errors. I found the paper to be hard to follow in places due to the unclear description of different models and time steps. The "Data and Methodology" section which describes the models/time steps analyzed was particularly difficult to follow; some of the paragraphs could be split and the writing could be more concise. Overall, the scientific approach and methods seem adequate, though I am new to this subject. The paper also presents clear conclusions that are a substantial contribution to scientific progress.

*Thank you very much for your time and your insightful review. We have carefully revised the manuscript in order to include your comments. We believe that this manuscript is substantially improved as a result of the revision.*

Below are other specific comments and suggestions:

Pg 1, line 13: Missing a word - "and generally are more skillful"

*RESPONSE: The missing word has been added as requested.*

Pg 1, line 16: Remove "the" before "number of consecutive dry days..."

*RESPONSE: It has been revised as requested.*

Pg 1, line 26: Add a comma after the reference.

*RESPONSE: It has been revised as requested.*

Pg 2, lines 13-19: To make the aims of this study more clear, it would be good to list or number them. There are 3 in the results section. It would be nice to see the same differentiation here.

*RESPONSE: Thanks for pointing this out. It has been revised as requested: "This study will conduct a comprehensive evaluation of the precipitation and temperature hindcasts at subseasonal timescales. Specifically, the aims of this study are to 1) assess the CFSv2 predictions for precipitation and temperature indices at different locations and seasons within the first 30 days, 2) compare weekly and fortnight forecast skill of CFSv2 at different lead times, and 3) evaluate the effects of MJO and ENSO on CFSv2 sub-seasonal forecast skill."*

Pg 3, lines 25-27: You refer to 0Z cycle (and different numbers of cycles). I am not familiar with this terminology, and am therefore confused when reading these sentences.

*RESPONSE: Thanks for pointing this out. 0Z cycle means the 0 UTC (Coordinated Universal Time) assimilation and forecast cycle. We have deleted these sentences and used a table and a figure instead to explain and compare those three CFSv2 hindcast configurations.*

Pg 3-4, lines 22-33, 1-11: This paragraph is confusing due to the large amount of information presented. The paragraph could be split into multiple paragraphs and reworded to clarify which time steps and models are being referred to.

*RESPONSE: We have split this paragraph into multiple paragraphs for clarification. Some of the sentences have been replaced by a table and a figure to explain and compare the three CFSv2 hindcast configurations. The paragraph has been reworded and updated accordingly.*

Pg 4, line 26: This could be the start of a new paragraph to separate the two different ideas.

*RESPONSE: We agree. It has been separated into two different paragraphs.*

Pg 5, line11: Replace comma with a semicolon or separate into two different sentences.

*RESPONSE: It has been revised as requested.*

Pg 8, line 23: Bold the title.

*RESPONSE: It has been revised as suggested.*

Pg 9, lines 26-28: This sentence is confusing and could be reworded to improve clarity.

*RESPONSE: This sentence has been modified as:*

*"This finding is important since the sub-seasonal forecasting information is valuable to many decision makers. In particular, sub-seasonal forecasts for frequency or duration of precipitation and temperature extremes can be directly tailored to different application needs."*

Pg 9, line 31: It is unclear which precipitation and temperature indices are being referred to with the word "these."

*RESPONSE: This sentence has been modified as "... those indices describing frequency or duration of precipitation and temperature extremes ..."*

Pg 10, lines 6-12: These sentences could be rewritten to be more concise.

*RESPONSE: This paragraph has been revised to be more concise and clearer.*

Pg 10, line 22: Figure 13 isn't used in the paper.

*RESPONSE: Thanks for pointing this out. It should be Figure 9 (Figure 12 in the revised manuscript) instead of 13. We have revised that in the manuscript.*

Pg 11, line 9: "Subseasonal" is spelled wrong and is also written differently than the other instances where the spelling was "sub-seasonal." This may be an issue throughout the paper and should be checked.

*RESPONSE: All the instances of "subseasonal" have been changed to "sub-seasonal" throughout the paper.*

---

## Author Comment (AC3) · 16 Oct 2016

Response to Reviewer #3:

General Comments and Suggestions:

The work presented here is interesting, and presents an opportunity for researchers to utilize sub-seasonal forecasts from the CFSv2 model. However, this manuscript would benefit from a thorough proofreading, as the authors' writing is difficult to follow throughout much of the paper. Overall, I believe the manuscript can be made more concise and direct, which I believe will improve the readability of the paper. I have noted some specific instances below, but I am sure it is not a comprehensive list of all the improvements that could be made to the manuscript.

*Thank you very much for your time and your insightful review. We have carefully revised the manuscript in order to include your comments. We believe that this manuscript is substantially improved as a result of the revision.*

On page 2, the authors state that, ": : :many extreme events and management decisions fall into sub-seasonal timescales: : :" I suppose I could use some examples. As the paper is currently, "sub-seasonal timescales" are on the order of 3-4 weeks. When I think of extreme events, I think of shorter term events such as flash flooding, tornadoes, extreme hail, wind, etc: : :, or I think of more persistent events such as drought or persistent floods. Many of the examples the authors provide in the second paragraph tend to be on the order of the seasonal timescale as defined by the authors on page 1. It would be beneficial if the authors could state what extreme events they specifically have in mind to be addressed by this research. Some of this is touched on near the end of the Discussion section by the authors, but could be more succinctly stated earlier.

*RESPONSE: We added following sentences to this paragraph:*

*"… For example, flash drought, heat wave, and cold wave are extreme events at sub-seasonal timescale. Sub-seasonal forecast information can be used for developing strategies for proactive natural disaster mitigation, which may be needed during those extreme events. … In this study, we aim to evaluate forecasting for precipitation and temperature derivatives, or indices that are associated with those extreme events, and decision-making at sub-seasonal timescale."*

At the end of the second paragraph on page 2, the authors state that the, ": : :derivitives or indices are directly associated with important events and decision making: : :". Similar to the previous comment, I do not believe the manuscript currently addresses this point.

*RESPONSE: In this study, we aim to evaluate forecasting for precipitation and temperature derivatives or indices that are associated with important events and decision-making at sub-seasonal timescale – in particular  the mean, frequency, duration, and intensity of precipitation and temperature at sub-seasonal timescale, such as the number of dry/wet days, number of cold/hot days, etc. We have noted this in the revised manuscript.*

Beginning on page 3 and continuing to page 4, the first paragraph of the Data and Methodology section is very confusing. A table comparing the differences between the 9-month, 45-day, and season runs would be beneficial. As it stands now, I believe this section assumes too much of the reader to interpret how NCEP runs the CFSv2 model.

*RESPONSE: We added a table and a figure to explain those three hindcast configurations. We have also updated the text in this section to make it more concise and direct.*

On page 4, the authors discuss an example and reference a "lead one" forecast. It appears they are referencing the 14-day forecast, but it's not clear.

*RESPONSE: This sentence has been clarified: "Taking the 14-day forecast for WetSpell in January as an example, the first (second) forecast lead time is the number of consecutive rainy days from January 1 to 14 (from January 15 to 28)."*

Equation 1 is not clear. The variable E is described as 1/3 the total number of forecasts, T. Why wouldn't the denominator simply be (T − T/3)? It sounds as if there is more to the variable E than is described.

*RESPONSE: The variable E is the expected number of categories forecast correctly just by chance. Since there are three forecast categories in this study, it is described as one third of the total number of forecasts. This has been clarified in the revised manuscript.*

With regards to the discussion of Figure 2 that starts near the end of page 6 and continues on to page 7, I am not sure why the authors do not include similar figures for MAM or SON, but do discuss the results. Wouldn't it be beneficial to include those figures? This comment applies to Figure 4 as well.

*RESPONSE: We agree that it would be beneficial to include those figures. The figures for MAM and SON have been added in the revised manuscript as suggested.*

Similarly, I'm not sure why the authors stress emphasizing the month of July in Figure 3. The authors state there is some difference in monthly spatial patterns, but without the other months, I do not have the proper context for the figure.

*RESPONSE: In the new figure, we have added the month of January as a context for the figure.*

With regards to section 3.3, it would be useful to have some sort of table or reference to see what periods of MJO and/or ENSO activity are being analyzed. The number of events considered could be limited enough that the HSS could be somewhat skewed. It would also be helpful if the authors discussed more clearly the impacts of active MJO and ENSO compared to the combined impacts of MJO and ENSO events. I think the authors begin to discuss this in the second paragraph on page 9, but do not offer enough insight on the particular point.

*RESPONSE: We have included a table showing the periods of MJO and ENSO events and point out that the limited number of events could somewhat skew the skill score conditioned on ENSO. We also discuss the combined effects of MJO and ENSO compared to the individual effects from either MJO or ENSO, suggesting a potential benefit of using MJO and ENSO information for sub-seasonal forecasts. Please see the details in section 3.3 of the revised manuscript.*

The first full paragraph on page 10, describing the role of the BM method is not clear. I would recommend the authors explain this conclusion more clearly, or simply remove the paragraph.

*RESPONSE: Thanks for pointing this out. We have simplified this paragraph by removing the sentences for describing the role of the BM method.*

On page 11, the authors state that forecast skill could be improved by simply having a larger ensemble. I am not convinced of that; a larger ensemble may not necessarily add useful information. It may be more appropriate to state that a sensitivity study on ensemble size could be performed to see if a larger ensemble does improve forecast skill.

*RESPONSE: We agree that a larger ensemble may not necessarily add useful information. We changed the original sentence to:*

*"Forecast skill could be potentially improved by having a larger ensemble size. A sensitivity study on ensemble size could be performed to assess whether a larger ensemble improves forecast skill."*

Page 1, lines 27-28: I'm not sure what is meant by, ": : :sub-seasonal timescale is beyond the memory of the atmospheric initial conditions: : :"

*RESPONSE: It means sub-seasonal timescale is sufficiently long that much of the memory of the atmospheric initial conditions is lost. We have clarified that in the revised manuscript.*

Page 5, the NCDC has since been renamed the National Centers for Environmental Information (NCEI)

*RESPONSE: It has been revised as suggested.*

Page 6, line 26 should read "Figure 2 shows"

*RESPONSE: It has been revised as suggested.*

There is no legend for Figure 5, so I am unsure which regions match to each particular color.

*RESPONSE: We have added a legend as suggested.*

Page 8, line 10: "temperally" should be "temporally"

*RESPONSE: It has been revised as suggested.*

Page 8, line 17: "reasonaly" should be "reasonably"

*RESPONSE: It has been revised as suggested.*

Page 9, line 23: "depending" should be "dependent"

*RESPONSE: It has been revised as suggested*

Page 10, line 4, "spatial" and "temporal" should be "spatially" and "temporally"

*RESPONSE: It has been revised as suggested.*

Page 10, line 5: "to note" should be "noting"

*RESPONSE: To address one of the comments raised above, this sentence has been deleted.*

Page 10, line 23: There is no Figure 13 included in the manuscript. To this point, referencing a figure in another article (Jones et al. 2011) is a bit confusing. I think I would just note how the results of this study compare to Jones et al., rather than to a specific figure.

*RESPONSE: Thanks for pointing this out. Figure 13 is a typo. It has been changed to Figure 12 in the revised manuscript. We agree it is a bit confusing to reference a figure in another article. We have changed that in the revised manuscript as suggested*

Page 11, line 8, "highlighted" should be "highlight"

*RESPONSE: It has been revised as suggested.*

Page 11, line 9, "subeasonal" should be "sub-seasonal"

*RESPONSE: It has been revised as suggested throughout the paper.*

---

## Editor Decision (ED1)

1.  The paper's treatment of the significance of the results is still less rigorous than I feel is appropriate for the forecasting topic.  For instance, even statements such as (p5 l1) "The HSS above 0 indicates that the forecasts have skill" (positive, negative?) are simplistic.  The skill estimate is certainly much stronger (ie significant) if the HSS has been calculated from a sample of 1000 obs-forecast pairs, compared to a sample of 10.   I recommend adding to the discussion of the skill score a small general discussion regarding uncertainty (due to sampling error) in the Heidke skill score estimates, but related to the samples sizes used in this paper – ie, 28.  It should be fairly straightforward to calculate a confidence interval given this sample size, which can then be referenced in the results discussion (Figures 3-10 excluding 5).  For instance, given sampling uncertainty with N=28 the HSS is significantly (positively, $p<0.05$ or $p<0.10$) skillful when it is greater than X (where X is greater than zero).

For the difference between two HSS, the CI may be more difficult to calculate analytically thus the use of the bootstrap is a convenient empirical approach.  Yet the authors use of the bootstrap on the sample of 6 mixed scores (3 months + 3 leads) is at least inadequate if not completely incorrect.  One strategy would be to generate 1000 trials of two 28-member non-skilled samples (perhaps by shuffling the obs-forecast pairs in time) and calculate score difference thresholds that are exceeded with a desired probability, purely by chance.  Another is to bootstrap similarly on the actual 28-member samples to assess the effect of their sample uncertainty on the difference in their scores.  Because the HSS is a widely-used metric in weather/climate forecasting, and it is likely a common challenge to assess whether one forecast is better than another, I expect examples can be found in the literature.  It is still a striking feature of figures 12-13 that adjacent pixels with different signals (eg -15, +10) or signals of zero are all found significant, when the underlying climate maps (NLDAS and CFSv2) are much smoother.  Despite an earlier request for analysis and discussion on this point, this issue remains unexplained.

Finally, the use of the significance calculations in the figures can be improved.  Rather than show the significance maps separately (eg in Figure 11), which makes it very difficult to match a pixel's significance and value, the example of skill masking at CPC could be followed – eg, http://www.cpc.ncep.noaa.gov/products/CFSv2/htmls/usPrece1SeaMask.html)

Before this paper can be accepted it will require a more rigorous and thoughtful treatment of uncertainties in the skill score estimates, referencing appropriate literature and clearly describing the approaches used to assign significance.

2.  In general, I feel the authors have adequately addressed the reviewers' comments, but I ask that the authors go back through the first round of comments to reconsider and upgrade any perfunctory and limited responses such as the one highlighted below.

*Reviewer:  Page 2, line 23-25: Please explain how GCM outputs can be used for daily or short-term forecasts seeing as they are uncorrelated to current meteorological conditions.*

*RESPONSE: Thanks for pointing this out. This is an overstatement. We have changed the sentence to "Coupled Atmosphere-Ocean General Circulation Models (GCMs) are used to make forecasts at multiple timescales."*

There are a number of interesting reasons why GCM outputs can and are being used for daily and short-term forecasts when they are initialized for weather and climate prediction, yet these are not discussed in the response or, more appropriately, the paper (as background, perhaps). The reviewer may be confused by the use of GCMs in free-running climate projection mode (where there is no correlation) rather than in operational forecasting mode, and here it would have been appropriate to make the distinction and to describe briefly the sources of GCM predictability at short-to-medium ranges (eg the initializations, inertial dynamics at different scales and in different components, etc.).

3. The writing still requires a careful proof-reading by an accomplished technical English-language writer.

---

## Author Response (AR2)

**Response to the Editor:**

*Thank you for your time in editing our manuscript and for your insightful suggestions. We have responded to the reviews point by point in the document attached. The revised manuscript with tracked marks is also attached behind the responses to the editor. We have submitted the revised manuscript as requested.*

1. The paper's treatment of the significance of the results is still less rigorous than I feel is appropriate for the forecasting topic. For instance, even statements such as (p5 l1) "The HSS above 0 indicates that the forecasts have skill" (positive, negative?) are simplistic. The skill estimate is certainly much stronger (ie significant) if the HSS has been calculated from a sample of 1000 obs-forecast pairs, compared to a sample of 10. I recommend adding to the discussion of the skill score a small general discussion regarding uncertainty (due to sampling error) in the Heidke skill score estimates, but related to the samples sizes used in this paper – ie, 28. It should be fairly straightforward to calculate a confidence interval given this sample size, which can then be referenced in the results discussion (Figures 3-10 excluding 5). For instance, given sampling uncertainty with N=28 the HSS is significantly (positively, $p<0.05$ or $p<0.10$) skillful when it is greater than X (where X is greater than zero).

*RESPONSE: Thanks for pointing this out. We used a bootstrapping technique to estimate the confidence interval of the HSS, and the average of the HSS over the United States as the test statistic. We have added the following statement to the methodology section of the revised manuscript: "Since the number of forecast-observation pairs is 28 for each point, the HSS estimation has considerable uncertainty given this relatively small sampling size. To quantify this uncertainty, a bootstrapping technique (Wilks 2011; Hamilton et al. 2012) was applied to resample 28 samples (3000 times with replacement) from the 28-year reforecast averaged over the CONUS. Then a number of 3000 HSS was calculated for constructing a distribution; with the confidence interval and significance level of the HSS estimated from this distribution. With this treatment of the HSS estimation uncertainty, we can determine that the HSS is significantly skillful when it is greater than a given significance level." We have calculated confidence intervals for different indices using this method and referenced that in the result discussion: "It is worth noting that given the sample size (N=28) used for calculating the HSS, the confidence interval of the HSS for each index is relatively wide. Based on the bootstrapping technique described earlier, the HSS was found to be significantly skillful (significantly above 0 at the 0.05 level) when it was greater than the number between 20 and 24, depending on the indices."*

For the difference between two HSS, the CI may be more difficult to calculate analytically thus the use of the bootstrap is a convenient empirical approach. Yet the authors use of the bootstrap on the sample of 6 mixed scores (3 months + 3 leads) is at least inadequate if not completely incorrect. One strategy would be to generate 1000 trials of two 28-member nonskilled samples (perhaps by shuffling the obs-forecast pairs in time) and calculate score difference thresholds that are exceeded with a desired probability, purely by chance. Another is to bootstrap similarly on the actual 28-member samples to assess the effect of their sample uncertainty on the difference in their scores. Because the HSS is a widely-used metric in weather/climate forecasting, and it is likely a common challenge to assess whether one forecast is better than another, I expect examples can be

found in the literature. It is still a striking feature of figures 12-13 that adjacent pixels with different signals (eg -15, +10) or signals of zero are all found significant, when the underlying climate maps (NLDAS and CFSv2) are much smoother. Despite an earlier request for analysis and discussion on this point, this issue remains unexplained.

*RESPONSE: Thanks for your comments and suggestions. We used a bootstrap technique for significance tests for the difference between the HSS during active MJO and the whole period. We resampled 28 samples (3000 times with replacement) from the 28-year reforecast averaged over the CONUS. All the 28 samples were used to calculate the HSS during the whole period. The subset of the 28 samples under active MJO events was used to calculate the conditional HSS. The difference between the HSS during the MJO and the all period was then calculated. Since the resampling was conducted 3000 times, a number of 3000 HSS differences was obtained for constructing a distribution and used to estimate the confidence interval and significance level of the HSS. Similar to Peng et al. (2013), the significance level estimated based on the average over the CONUS was applied to test the local significance for each grid point over the region. The results have been updated in Figures 12 and 13 and relevant texts in the revised manuscript.*

Finally, the use of the significance calculations in the figures can be improved. Rather than show the significance maps separately (eg in Figure 11), which makes it very difficult to match a pixel's significance and value, the example of skill masking at CPC could be followed – eg, http://www.cpc.ncep.noaa.gov/products/CFSv2/htmls/usPrece1SeaMask.html)

*RESPONSE: Thanks for your suggestions. We have used mask to improve the use of the significance calculations in the Figures 12 and 13.*

Before this paper can be accepted it will require a more rigorous and thoughtful treatment of uncertainties in the skill score estimates, referencing appropriate literature and clearly describing the approaches used to assign significance.

*RESPONSE: Thanks for your comments. We have addressed those issues in the revised manuscript. Please refer to the responses to the comments above.*

2. In general, I feel the authors have adequately addressed the reviewers' comments, but I ask that the authors go back through the first round of comments to reconsider and upgrade any perfunctory and limited responses such as the one highlighted below.

*Reviewer: Page 2, line 23-25: Please explain how GCM outputs can be used for daily or short-term forecasts seeing as they are uncorrelated to current meteorological conditions.*

*RESPONSE: Thanks for pointing this out. This is an overstatement. We have changed the sentence to "Coupled Atmosphere-Ocean General Circulation Models (GCMs) are used to make forecasts at multiple timescales."*

There are a number of interesting reasons why GCM outputs can and are being used for daily and short-term forecasts when they are initialized for weather and climate prediction, yet these are not discussed in the response or, more appropriately, the paper (as background, perhaps).

The reviewer may be confused by the use of GCMs in free-running climate projection mode (where there is no correlation) rather than in operational forecasting mode, and here it would have been appropriate to make the distinction and to describe briefly the sources of GCM predictability at short-to-medium ranges (eg the initializations, inertial dynamics at different scales and in different components, etc.).

*RESPONSE: Thanks for your comments. We again went through the reviewers' comments and found that the explanation for the use of GCMs outputs is not sufficient. We have added the following to clarify this in the revised manuscript:*

*"Coupled Atmosphere–Ocean General Circulation Models (GCMs) are used to make forecasts at multiple timescales, from medium-range weather forecasting, seasonal climate predictions, and long-term climate projections. The reason GCMs can be used as operational models at these time scales is due to the predictability from different sources, such as initial conditions from the atmosphere, inertial dynamics from soil moisture and sea surface temperature, etc."*

3. The writing still requires a careful proof-reading by an accomplished technical English language writer.

*RESPONSE: Thanks for your comments. We have conducted a careful proof-reading to improve the writing of this manuscript.*

[revised manuscript text omitted]